# Distortions to Passage of Time Judgements (POTJ) due to virtual threat are predicted by autonomic activation

Stephen H. Fairclough [1‡*], Christopher Baker [2‡], Ruth Ogden [1‡], Rachel Barnes [1], Jessica Toothill [1]

1 School of Psychology, Liverpool John Moores University, Liverpool, United Kingdom, 2 School of Computer Science and Mathematics, Liverpool John Moores University, Liverpool, United Kingdom

‡ SHF, CB and RO are joint senior authors on this work.
* s.fairclough@ljmu.ac.uk

**Data Availability Statement:** The data are available at the LJMU Open Data Repository (https://opendata.ljmu.ac.uk), and the DOI for the data is as

## Abstract

Humans' sense of the passage of time is subjective and determined by psychophysiological responses to the environment. The passing of time has been perceived to significantly slow in stressful situations, such as accidents or virtual threats. The current study will explore distortions in the perception of passage of time when threat is simulated using virtual reality (VR). 44 participants negotiated a large (13.6 × 8.4 m) virtual environment designed to maximise the realism of a physical threat by exposing participants to a virtual height of 200m. Subjective perception of passage of time and time estimation were used as independent variables, whereas the movement of participants, and ambulatory psychophysiology, e.g., electrodermal activity (EDA), heart rate, served as dependent variables. The independent variables were examined in relation to the dependent variables through a regression analysis, which allowed for the identification of the specific weight of each variable. Our analyses revealed that passage of time was perceived to decrease (i.e., time slowed down) for those participants who exhibited the higher levels of skin conductance (SCL). It is argued that this finding can be explained by individual differences in self-regulatory strategies during the task and the effectiveness of VR as means to simulate threat.

## Introduction

Our sense of the speed of the passage of time is highly subjective and vulnerable to distortion by events in our environment [1–7]. As a result, time can often feel like it is passing more quickly or slowly than normal [8]. Whilst small distortions to time are common during everyday life, e.g., meetings passing slowly due to boredom, the most profound temporal distortions occur when we experience extreme changes in emotional and cognitive states [9]. For example, analysis of accounts of near-death experiences during car crashes and natural disasters, reveal a near universal reporting of the subjective sensation of a significant slowing of the passage of time [10].

follows: https://doi.org/10.24377/LJMU.d.00000195.

**Funding:** Funded by Experimental Psychology Society The funders had no role in study design, data collection/analysis, the decision to publish, or the preparation of the manuscript.

**Competing interests:** The authors have declared that no competing interests exist.

Extreme changes in psychological states may distort the passage of time by disrupting homeostasis. The model of time perception [11] proposed that extremes of psychophysiological arousal distort the passage of time via top-down control on homeostatic regulation exerted by the anterior insula cortex (AIC). During exposure to threat, the right AIC is highly activated and sympathetic nervous system (SNS) activity increases. Critically, the AIC is also activated during temporal processing [11], leading Craig to argue that antagonistic activation of the AIC during homeostatic regulation *and* temporal processing is a causal factor in distortions to the passage of time. During periods of high emotional arousal, right-side AIC activity increases the number of emotional-temporal units processed in the brain, leading to a dilation of time and the subjective sensation of time passing more slowly than usual.

The basic predictions from this homeostatic model are supported by laboratory-based studies of the processing of short (<1 minute) durations. Piovesan et al., (2019) [12] asked participants to estimate the duration of electro-cutaneous stimuli administered at levels to produce: no pain, low pain and high levels of pain. Differences in passage of time estimation were predicted by the changes in skin conductance level (SCL) between the experimental conditions, suggesting that temporal distortions were determined by activation of the sympathetic nervous system (SNS). Similar findings were observed when threat was induced by exposing participants to images from the International Affective Picture System, Ogden et al., (2019) [13] found that SNS reactivity to the high threat images was predictive of distortions to the perceived length of the high threat images. Critically, changes in SNS activation were only predictive of temporal distortion when participants viewed negative images, suggesting that SNS activity is only associated with temporal judgements in those specific instances when the participant experienced a degree of threat and negative emotion. Furthermore, the only existing study exploring the relationship between psychophysiological measures and the passage of time over longer durations (1 hour) during real-world activity also reported that SNS reactivity predicted of distortions to the passage of time [14].

Research on the association between threat, autonomic activation and time distortion is limited by several obstacles. In the first instance, it is unethical to place human participant in actual physical jeopardy to induce threat. As described in the last paragraph, experimental pain and emotional induction present a mild form of threat for participants, but the controlled artificiality of these protocols limits the verisimilitude of threat experienced by participants. However, Virtual Reality (VR) offers a method for inducing threat and those strong emotional responses associated with the simulation of physical jeopardy. VR can also create digital worlds that maintain experimental rigour, permit standardisation and provide ecological validity [15] and this technique has been used extensively as an emotion induction paradigm, see reviews by Bernardo et al (2021) [16], Somarathna et al (2023) [17], and Kako et al [18].

For all its technical sophistry, it remains questionable whether VR can successfully provoke the intense emotions experienced during moments of physical jeopardy and a number of experiential criteria must be met. It is important for the technology to combine high-resolution visuals, accurate motion tracking, and dynamic interactivity to achieve a sense of immersion [19]. Although the exact definition and modalities required to define presence remain under debate [20], they can be grouped into three categories: the technical capacity of the system to stimulate the somatosensory systems of the user [21], the presentation of a plausible and consistent virtual world with coherent physical laws [22] and a sense of agency, i.e., users feel their actions have consequence to the VE and to themselves [23].

Several different VEs have been created to induce threat in participants that involve a simulation of height. The original "pit" scenario [24, 25] required a participant to navigate towards a virtual precipice. This experimental paradigm was adapted to increase affect and incorporate more complex stimuli and sensorimotor contingencies [26–28] and co-opted for use in

psychophysiological [29, 30] and neurophysiological studies [31, 32]. Enhancements in graphical fidelity, the use of props, improved tracking, and room-scale capabilities have significantly increased the immersive qualities of these simulations [33]. A combination of immersion, presence and agency reinforces negative emotions (e.g., anxiety, discomfort) and a tendency towards avoidant behaviour in participants when they are exposed to height and the risk of a fall [34]. A genuine sense of threat adds to the ecological validity of the virtual scenario and the resultant data captured from behavioural response.

The current study utilised a large, room-scale virtual environment (VE) designed to induce anxiety and fear by exposing participants to the threat and the experience of a virtual fall. It was hypothesised that participants experience of the passage of time during the ice-walk task would be predicted by their physiological responses to the task. Specifically, based on the homeostatic model of time, it was anticipated that a slowing of the passage of time would be associated with increases in heart rate and SCL.

## Materials and methods

### Participants

Forty-four participants (22 female) were recruited to the study from a university population of undergraduates and postgraduates. The mean age of the participants was 24.72 yrs (SD = 6.75 yrs). The only criterion for inclusion was that participants must be aged over 18 years and able to walk without assistance.

### Ethics statement

The experimental protocol was approved by the Liverpool John Moores University Research Ethics Committee prior to data collection (Ref: 23/PSY/007), which operates within guidelines from the UK Research Integrity Office Code of Practice for Research and in accordance with the Declaration of Helsinki. Participants were fully informed prior to agreement to take part in the study and provided written consent.

### Virtual environment

Participants were required to negotiate a path across a large grid of ice blocks suspended in the air at a virtual height of 200m. The layout of the VE is illustrated in Fig 1 and the participants' view shown in Fig 2. Participants were required to follow a route from section 1 to section 9 (Fig 1). The end goal of the VE was represented by a door that must be activated by hand to leave the VE (Fig 2). The VE occupied a physical space of 13.6 × 8.4 m, and each individual block shown in Fig 1 was approximately 70 x 70cm.

Participants interacted with the blocks via foot movements, which was achieved by attaching sensors to participants' feet in addition to conventional handheld trackers. Foot sensors allowed participants to interact with ice blocks in two ways: (1) a one-footed movement to test the block before stepping onto it with both feet, and (2) a two-feet movement in which participants moved fully to the block and stood on it with both feet.

The grid of ice blocks contained three types of blocks. If the block was Solid (green in Fig 1), it would support the weight of the participant and did not change appearance when activated with either one-foot or two-feet interaction. Crack blocks (blue in Fig 1) would also support the weight of the participant but any interaction caused a change of colour from translucent to blue accompanied by a cracking sound effect 500ms after activation. Fall blocks (red in Fig 1) would not support the weight of participants. A Fall block behaved in the same fashion as Crack blocks during a one-foot interaction, i.e., the block would change colour and

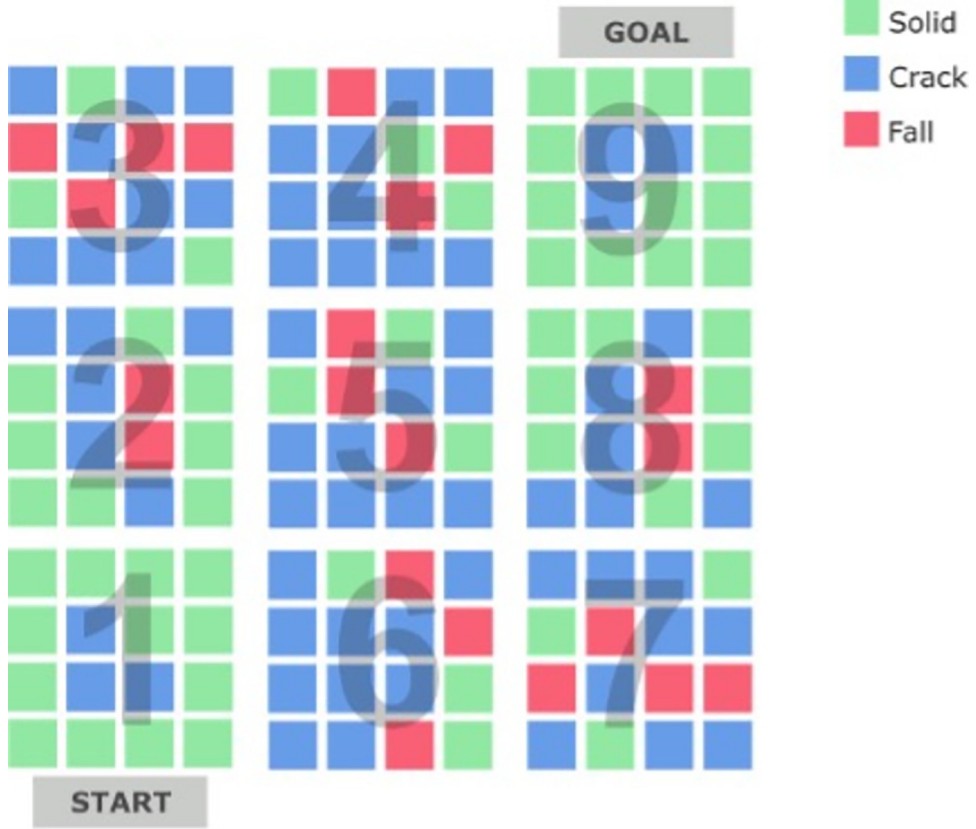

**Fig 1. Layout of the virtual environment.**

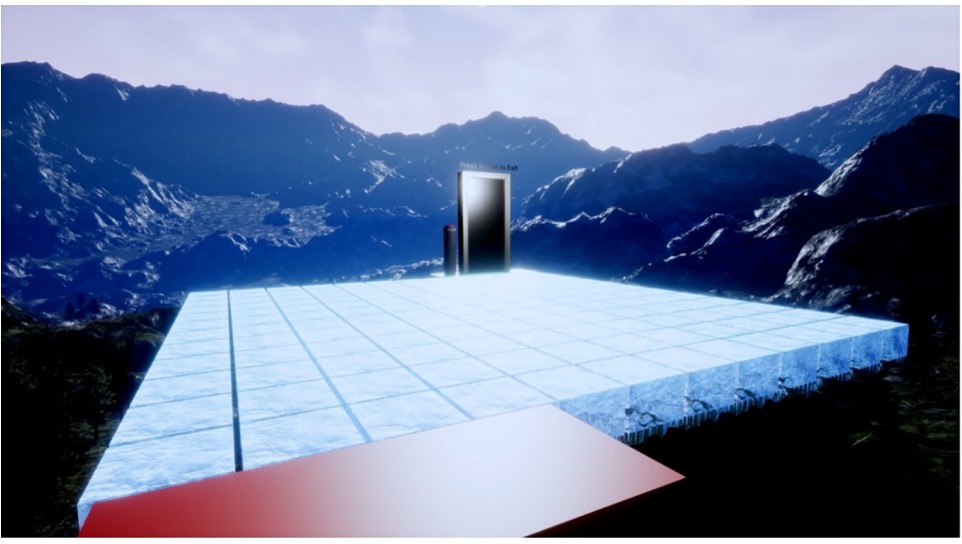

**Fig 2. Participants' view of the virtual environment.**

make a cracking sound, but any subsequent two-feet activation to the block triggered a shattering sound effect after 500ms whereupon the block would disintegrate, and participants experienced a virtual fall. Therefore, participants learned to use one-footed movements to test each block to identify Crack blocks, which could also be the Fall blocks that they should avoid.

The composition of blocks within the VE, specifically the inclusion of Crack and Fall blocks, was configured to manipulate the level of threat experienced by participants (Fig 1). From section 1 to section 3 (S1:S3), the number of Crack and Fall blocks increases in a linear fashion to create an experience of increasing threat. In the middle sections (S4:S6), the level of threat is maximal as most blocks are Crack and Fall blocks, making the experience of a virtual fall difficult to avoid and forcing participants to make risky two-footed movements to cracked blocks that could lead to a fall. In the final 'leg' of the VE (S7:S9), the array of blocks from S1: S3 is mirrored precisely so that there is a reduction in the number of Crack and Fall threat blocks in a linear fashion. S7:S9 is configured in this way to assess the extent to which participants adjust their strategy to adapt to a reduction of threat.

As participants moved forward through the VE, the previous two rows were removed from behind participants to remove any opportunity for "backtracking", i.e., participants must keep moving forward. In the event of a fall, participants were returned to the block that they occupied before a fall and a gap would appear in the grid to indicate the former position of the Fall block. A video of a participant negotiating the VE is included in the S1 Video.

## Virtual reality system

Participants wore an HTC Vive head-mounted display (HMD) to experience a bespoke VE created using Unreal Engine 4.27. Each participant held two hand controllers and two trackers were attached to their feet. Hand and feet positions were represented as robotic hands and white luminous outline respectively in the VE. All assets were purpose built for the study. The VE was rendered on a desktop PC with custom C++ code to capture interactions with blocks and recorded timings.

## Temporal perception

Upon completion of the ice-block walk, participants were asked 'how quickly did you feel that the task passed' and they completed a 10-point Passage of Time Judgement scale, where 1 represented time passing very slowly, 5 represented a mid-point (i.e., time passing normally) and 10 represented time passing very quickly. Participants were also asked to provide an estimate in minutes and seconds of how long it has taken for them to complete the ice-block walk, which was subsequently converted into seconds for analysis.

**Psychophysiology.** Skin Conductance Level (SCL) was recorded at 2000 Hz via the Bionomadix ambulatory psychophysiology system (BIOPAC). This psychophysiological measure captures the electrical resistance of the non-glabrous skin and is associated with activation of the sympathetic nervous system. SCL data were collected from the index finger and second digit of the non-dominant hand and processed in python using the cvxEDA Convex Optimisation to Electrodermal Activity Processing function to extract the phasic and tonic components of the signal. An electrocardiogram (ECG) was recorded at 2000 Hz via the Bionomadix system with three disposable electrodes placed on the left and right sides of the collarbone and the lower left rib cage. The mean heart rate (HR) was processed in python via the neurokit2 library and the nk.ecg peaks function. For behavioural data, Time taken in Sections of the VE, one-foot and two-footed movements, markers were generated by the Unreal Engine VE to specify when a participant began and ended the study. These markers were activated either automatically when the study began or via trigger volumes within the VE i.e. touching the virtual button

at the end of the task. Markers where also generated via the worn foot trackers during differing interactions, i.e., individually for one-footed movements when are participants interacted with one-foot on a virtual block, or in timed beginning and end pairs for two footed decision events i.e. when both feet step on a block and a corresponding end event when both feet stepped off a block.

## Procedure

Participants arrived at the laboratory, where they read a Participant Information Sheet (written instructions about the task and the VE), were given an opportunity to ask any additional questions of the experimenter, and subsequently provided written consent. The SCL sensors were taped to the second phalanx of fingers on the non-dominant hand and disposable ECG electrodes attached to both sides of the collarbone and the rib cage. Signal quality of the psychophysiological data was checked. Participants were subsequently fitted with the Vive Tracker sensors on their feet, which were attached to their shoes via velcro-straps. The HMD were placed over the head, adjusted and checked for comfort, and participants received the handheld controllers. The baseline VE phase was activated and participants were required to stand in a neutral grey toned environment designed to provide no sensory stimulation for a period of 3 minutes. Participants were instructed to relax during the baseline period. The task VE was activated after this phase and participants stood at virtual height on the starting platform before the first section of 16 ice-blocks (Fig 1). Participants could progress at their own speed, thus introducing autonomy and variability in emotional and physiological responses, which was anticipated to enhance the ecological validity of the studySCX. As they advanced and reached the end row of each the next Section would animate upwards. This design decision prevented rapid advancement and ensured progress into each successive Section preventing advancement to the goal via a direct route, e.g. diagonally across the grid. After advancing through Section 9 participants activated the goal doorway and proceeded through it. Participants then completed the Passage of Time Judgement scale and were asked to provide a time estimate. They subsequently had the VR apparatus and psychophysiological sensors removed, were thanked for their time and debriefed.

## Hypotheses and statistical analyses

A number of multiple linear regression models were created to explore the association between subtypes of trait impulsivity (independent variables) and various dependent variables, from behavioural outcomes (e.g., time duration, mean time spent on block, interactions with blocks) to psychophysiological measures (e.g., mean skin conductance level, heart rate). In those instances where we assumed that behavioural outcomes (e.g., frequency of falls, speed of movement) would affect psychophysiology, we included behaviour in the model as an independent variable. The linear regression models were generated using SPSS v.28 (IBM). For all models, normality of residuals was checked and the relationship between normalised residuals and predicted values was visually explored. If any normalised residual or predicted value was greater than 3 or less than -3, that participant was deemed to be outlier and excluded from the analysis.

With respect to statistical power, we included two multiple regression models in our analyses, both of which contained 5 predictors. Sample sizes for both models were calculated using G*Power. We used the setting for 'Linear Multiple Regression: Fixed model, $R^2$ deviation from zero' from G*power where the alpha level was set to 0.05 and Power was set to 0.80. With respect to effect size, we anticipated a large effect size given the strong emotional responses to

**Table 1. Descriptive statistics for all variables related to time.** Total time = time taken to complete ice-walk task, Time Estimate = participants' estimation of time taken to complete ice-walk task, POTJ = Passage of Time Judgement on 10-pt scale. All figures (except for POTJ) are presented in seconds (N = 44).

| Variable | Mean | Std. Dev | Min | Max |
|---|---|---|---|---|
| Total time (sec) | 174.76 | 108.38 | 52.06 | 713.31 |
| Time Estimate (sec) | 265.51 | 178.56 | 60 | 900 |
| POTJ | 6.68 | 2.52 | 1.2 | 10 |

the VE observed in previous work, we decided to set the effect size to 0.35. On this basis, we subsequently calculated the minimal sample size to be N = 43.

## Results

Participants could complete the ice-walk task at their own pace and were subsequently asked to indicate the duration of the task as well as a subjective estimate of whether time passed more slowly or quickly than usual, i.e., Passage of Time Judgement (POTJ). Descriptive statistics are presented for all data related to time duration and time judgements in Table 1 below.

Inspection of the descriptive statistics indicated clearly that participants tended to overestimate the actual duration of the ice-walk task; on average, the duration of the task was overestimated by 90.75s. Two multiple regression models were created using independent variables from behaviour and psychophysiology to predict POTJ and time estimate data. These models included the total time taken to perform the ice-walk task, the number of two-footed movements made during the ice-walk, and the frequency of falls as behavioural variables. With respect to psychophysiology, baselined means of skin conductance level (SCL) and heart rate (HR) were included in the regression models, i.e., mean activation of SCL and HR during the ice-walk task minus mean activation of SCL and HR during baseline period.

Table 2 presents a correlation coefficient matrix of all variables that were included in the two regression models.

The first regression model utilised all five independent variables to predict scores on the Passage of Time Judgement (POTJ) scale. The data were inspected for outliers (i.e., >3 or <-3) with respect to standardised predicted and residual values, this exercise led to the rejection of data from 5 participants, reducing the total N for this analysis to 37. The POTJ model was found to be statistically significant [$F_{(5,31}$ = 2.85, p = 0.03] with a $R^2$ value of 0.32 (Adj. $R^2$ = 0.28). Coefficients for the model are presented in Table 3.

**Table 2. Correlation matrix of all variables (N = 43).** POTJ = Passage of Time Judgement, Time_Est = estimate of task duration, Total_Time = task duration, Falls = total number of falls, Moves = total number of two-feet movements, SCL = baselined skin conductance level, HR = baselined heart rate. * = sig at p < .05, ** = sig at p < .01.

| | POTJ | Time_Est | Total_Time | Falls | Moves | SCL |
|---|---|---|---|---|---|---|
| Time_Est | -.503 ** | | | | | |
| Total_Time | -.202 | .246 | | | | |
| Falls | .263 | -.347 * | -.010 | | | |
| Moves | .096 | .349 * | .274 | .108 | | |
| SCL | -.301 * | .197 | -.038 | .043 | .146 | |
| HR | -.045 | -.024 | .002 | .346 * | .181 | .075 |

**Table 3. Coefficient table of all independent variables in the POTJ model: Std. Beta = Standardised Beta Coefficient, t = t-value, Partial-r = partial correlation.**

| IV | Std. Beta | t | Sig. | Partial r | Tolerance |
|---|---|---|---|---|---|
| Total_Time | -0.204 | -1.310 | .200 | -.229 | 0.910 |
| Moves | 0.292 | 1.807 | .080 | .309 | 0.848 |
| **Falls** | **0.411** | **2.463** | **.020** | **.405** | **0.793** |
| **SCL** | **-0.413** | **-2.677** | **.012** | **-.433** | **0.926** |
| HR | -0.328 | -1.924 | .064 | -.327 | 0.762 |

The scatterplots for both significant coefficients are illustrated in Fig 3. The model revealed significant positive associations between POTJ rating and falls, i.e., greater frequency of falls was associated with a feeling that time passed more quickly than usual, which may be

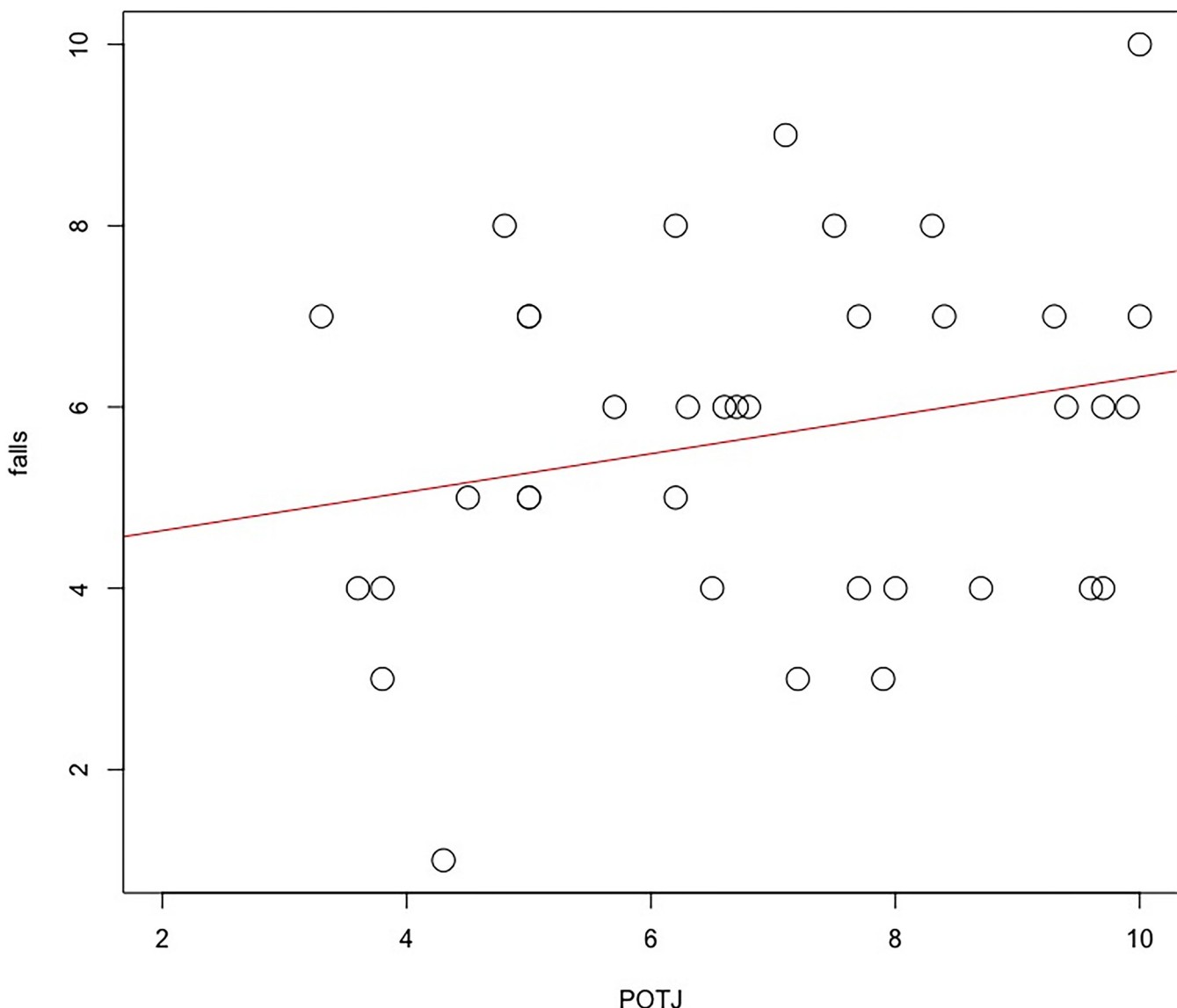

**Fig 3. Scatterplot of frequency of falls with Passage of Time Judgement rating scale (1 = time passing more slowly than usual, 10 = time passing more quickly than usual) (N = 37).**

**Table 4. Coefficient table of all independent variables in the time estimation model: Std. Beta = Standardised Beta Coefficient, t = t-value, Partial-r = partial correlation.**

| IV | Std. Beta | t | Sig. | Partial r | Tolerance |
|---|---|---|---|---|---|
| Total_Time | 0.358 | 2.345 | .025 | .378 | 0.926 |
| Falls | -0.311 | -1.996 | .054 | -.328 | 0.886 |
| Movements | 0.196 | 1.253 | .219 | .213 | 0.885 |
| SCL | 0.028 | 0.190 | .850 | .033 | 0.988 |
| HR | 0.193 | 1.232 | .227 | .210 | 0.878 |

attributable to increased emotional arousal at those moments that distorts self-awareness and time perception. By contrast, the negative association between baselined skin conductance level and POTJ rating indicated that time was perceived to pass more slowly for participants who exhibited higher levels of skin conductance level.

An identical regression model was created using time estimation, i.e., the duration of the ice-walk in minutes and seconds as estimated by participants, as the dependent variable. As in the previous analysis, data were inspected for outliers and 3 participants were removed (N = 39) from the regression. The resulting model had a $R^2$ value of 0.29 (Adj. $R^2$ = 0.24) and was statistically significant [F(5,33) = 2.69, p = 0.04]. The coefficients for this model are presented in Table 4.

The analysis of time estimate data revealed an expected positive association between actual time duration of the task and estimated time. We also found a significant negative relationship between the number of falls and estimations of time, i.e., participants who experienced a greater number of falls underestimated the actual duration of the task.

## Discussion

The analysis of Passage of Time Judgement (POTJ) revealed that time passing faster was associated with a greater number of falls (Table 3 and Fig 3). The same model indicated that participants who exhibited higher autonomic activation (greater SCL change from baseline) perceived time to pass more slowly (Table 3 and Fig 4). The latter result is supported by heart rate data, which showed the same relationship, however, this inverse association fell just outside statistical significance (Table 3). The influence of autonomic activation on time judgement was specific to the POTJ variable, neither psychophysiological variable was a significant predictor of estimated time (Table 4). Estimated time was significantly and unsurprisingly predicted by actual time duration and negatively predicted by the number of falls in this model (Table 4), i.e., lower time estimate for participants who fell more frequently.

There is a precedence in the literature for participants to overestimate the passage of time when experiencing emotions associated with high activation, such as fear and anxiety [12, 13]. The current study provides further support for this position and the broad conclusion, suggested in the model by Craig et al. (2009), that how an individual responds physiologically to threat exerts a significant influence on temporal experience.

The range of individual differences in SCL reactivity that we observed in the current study (Fig 3) may be driven by different factors. In the first instance, participants were granted a high degree of autonomy within the VE, specifically, both speed and direction of movement were self-determined, as was the decision to 'check' each block with one foot prior to each two-feet movement. This agentic aspect of the task elicited a variety of self-regulatory strategies. For example, some participants performed the task in a thoughtful, reflective fashion, e.g., taking longer between decisions, performing frequent checks prior to movement, trying to remain calm and focused with an internal locus of control. Others approached the ice-walk

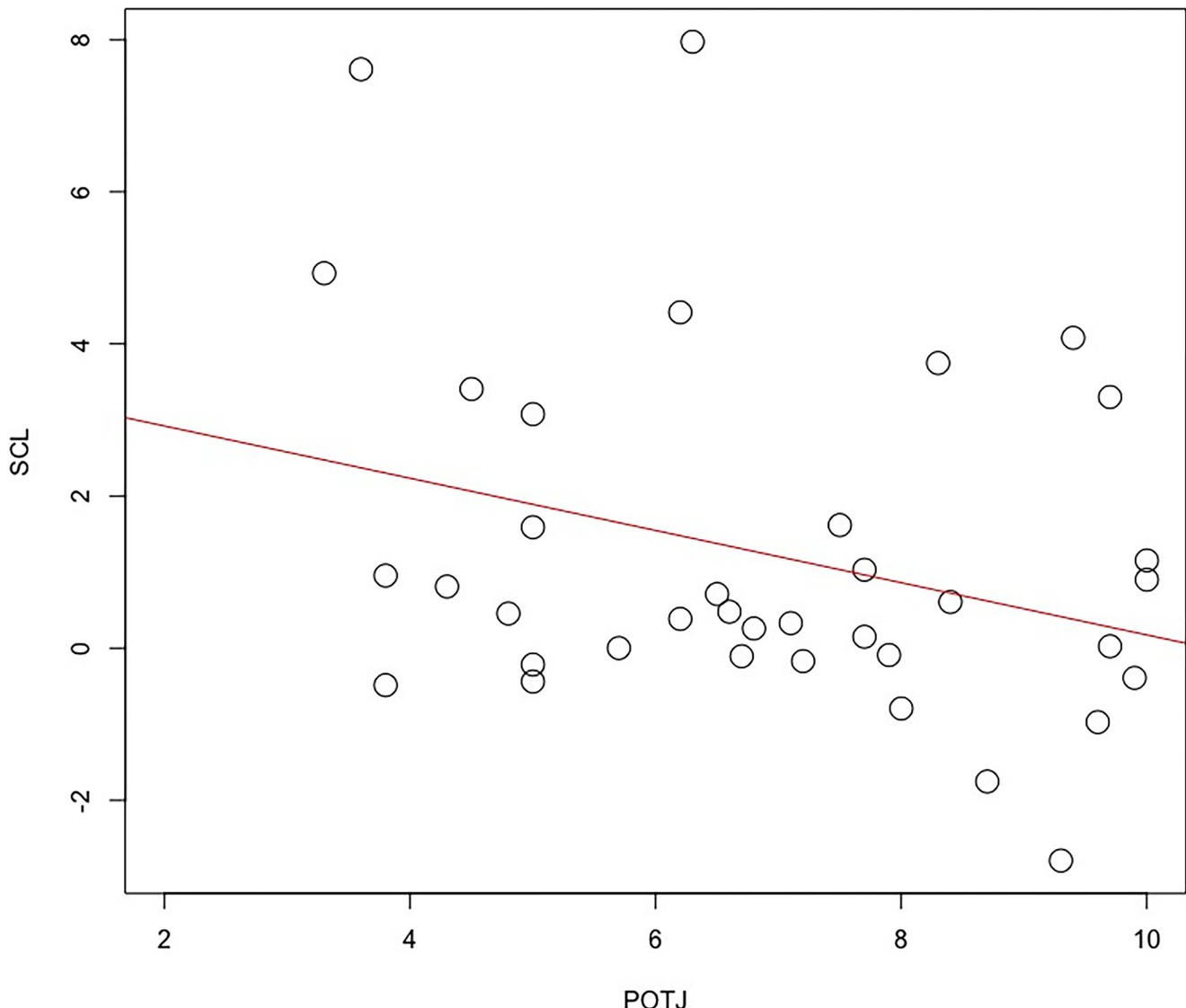

**Fig 4. Scatterplot of baselined skin conductance level in MicroSiemens (SCL) with Passage of Time Judgement rating scale (1 = time passing more slowly than usual, 10 = time passing more quickly than usual) (N = 37).**

from a more fatalistic perspective, relying on chance to avoid falls and moving without few checks, this strategy involved adopting an external locus of control and experiencing increased anxiety/fear–and exhibiting greater SCL reactivity as a direct result.

Our results may also be influenced by individual differences with respect to how participants responded to the immersive properties of the VE itself. The ice-walk task was designed to induce threat by: (1) conveying a realistic illusion of height from a visual perspective, (2) mapping the spatial properties of the virtual space directly onto an actual physical space, and (3) optimising the illusion of body ownership by matching hand/foot movements of avatar with hand/foot movements of the participant [19]. This sense of 'being there' [22] was crucial to eliciting threat and strong emotional responses to virtual height [31, 32, 35–37]. However, the practical effectiveness of this illusion differs across individuals. Some will be predisposed towards higher autonomic reactivity in the VE due to existing personality traits, such as

anxiety or vertigo [32, 36]. Participants' physiological responses may also vary due to differences in the perception of realism when experiencing a virtual threat [31]. Unfortunately, neither trait measures of anxiety/vertigo or subjective self-report data on presence were collected in the current study. Future studies should consider including measures of anxiety and presence to better account for the variability in autonomic responses observed in the current study.

There were several other limitations in the study, for example, the number of participants (N = 44) could produce preliminary results, especially when one considers how this number was reduced due to the presence of outliers in the sample; in addition, the sample were limited in terms of their demographic profile. It is recommended that the study is replicated with a larger sample (and perhaps broader population) in order to assess the generalisability of our results. While the VE was large, immersive and realistic, there was no physical jeopardy associated with the task, which begs a question of whether the visual simulation of threat is sufficient to elicit a realistic response to danger without any consequences for the individual. This issue is difficult to resolve while maintaining the necessary ethical standards of experimental work with human participants, but does leave unanswered questions about the generalisability of our findings to real-world situations.

Our findings inform the debate about the extent to which passage of time judgements are distinct from other forms of duration judgment [3], precisely how humans accomplish passage of time judgements remains unclear. There is evidence that subjective changes in subjective emotional arousal specifically affect real world passage of time judgements but not retrospective judgements [3], a position supported by the finding that changes in low-level visual complexity affects passage of time judgements and duration judgments separably [38]. This distinction is supported by the findings of the current study which shows differential effects of autonomic activity on passage of time judgements that was not replicated for retrospective estimates of duration. The absence of an effect of autonomic activation on retrospective estimates supports suggestions that these types of judgement are primarily based on mnemonic processes [39].

Our finding that faster passage of time was associated with a greater number of falls was unexpected. It is possible that the experience of a virtual fall was both attention-demanding, activating and negatively valenced, which could momentarily diminish the process of self-awareness, which is fundamental to our experience of time; therefore, falling more frequently repeatedly distracted from cues that inform temporal judgement. It is also possible that an increased number of falls represented individual differences in perceived task difficulty; in other words, participants who fell frequently perceived successful negotiation of the ice-block task to be more challenging, which created a time dilation effect [40]. A final possibility is that the experience of speed during each virtual fall altered participants' subjective experience of the passage of time. A theory of magnitude [41] suggests the existence of a shared processing system in the parietal cortex, responsible for various forms of magnitude processing e.g. time, space, numerosity and size. Within this system, magnitude representations are thought to be monotonically mapped, so that changes in one magnitude domain (e.g., increased time) correspond with changes in other domains (e.g., increased space). It is possible to observe cross-domain interference within this system whereby magnitude information from one domain (e.g. space) influence the experience of another magnitude domain (e.g. time) [41, 42].

To summarise, we reported that individual variation of autonomic reactivity was predictive of passage of time judgement. This relationship may be driven by individual differences in self-regulatory strategy and the perceived realism of the virtual threat. Individual variation in SCL reflected the extent to which participants experienced genuine threat in the VE and the emotional consequences of genuine threat, and only participants who responded

physiologically to the VE reported a subjective slowing in the passage of the time. Further research is required to investigate this hypothesis further by including measures of immersion/ presence as well as relevant personality traits, alongside the measures of behaviour and psycho-physiology that were used in the current study.

The results of the current study should be regarded as a tentative first step in exploring how autonomic reactivity relates to time judgements in the presence of threat. While the VE utilised in the study has been previously used to induce threat [30], it permits a high degree of participant autonomy, which inevitably increases variation in the data and the complexity of observed behaviour. The use of immersive virtual environments represents a methodological innovation that can be extended to other applications, such as training and rehabilitation. Specifically, VR is increasingly used in training military or emergency services personnel to perform under conditions of high-stress, while realism is important, personal autonomy with the VE induces individual variability in strategies and performance that permit an assessment of the individual to function under conditions of duress. Our work also suggests that distortions to passage of time judgements may have methodological significance for the assessment of immersion and presence when using VR. The incorporation of time judgements into the assessment of user experience may be especially for therapeutic applications where distortions to temporal experience are part of the psychological experience being simulated by the technology.

## Supporting information

**S1 Video. A description of the ice walk task and demonstration of the virtual environment used in the study.**
(MP4)

## Author Contributions

**Conceptualization:** Stephen H. Fairclough, Christopher Baker, Ruth Ogden.

**Data curation:** Stephen H. Fairclough, Christopher Baker.

**Formal analysis:** Stephen H. Fairclough, Christopher Baker.

**Funding acquisition:** Ruth Ogden.

**Investigation:** Stephen H. Fairclough, Christopher Baker, Ruth Ogden, Rachel Barnes, Jessica Toothill.

**Methodology:** Stephen H. Fairclough, Christopher Baker, Ruth Ogden.

**Software:** Christopher Baker.

**Visualization:** Stephen H. Fairclough.

**Writing – original draft:** Stephen H. Fairclough, Christopher Baker.

**Writing – review & editing:** Stephen H. Fairclough, Christopher Baker, Ruth Ogden.

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
