## [Decision Letter · Decision Letter 0]

7 Oct 2024

PONE-D-24-39788

DISTORTIONS TO PASSAGE OF TIME JUDGEMENTS DUE TO VIRTUAL THREAT ARE PREDICTED BY AUTONOMIC ACTIVATION

PLOS ONE

Dear Dr. Fairclough,

Thank you for submitting your manuscript to PLOS ONE. After careful consideration, we feel that it has merit but  needs minor revision PLOS ONE’s publication criteria as it currently stands. Therefore, we invite you to submit a revised version of the manuscript that addresses the points raised during the review process.

We look forward to receiving your revised manuscript.

Kind regards,

Angelo Rodio

Academic Editor

PLOS ONE

Journal Requirements:

2. Thank you for stating the following financial disclosure: “Funded by Experimental Psychology Society” Please state what role the funders took in the study. If the funders had no role, please state: "The funders had no role in study design, data collection and analysis, decision to publish, or preparation of the manuscript." If this statement is not correct you must amend it as needed. Please include this amended Role of Funder statement in your cover letter; we will change the online submission form on your behalf.

3. We note that you have indicated that there are restrictions to data sharing for this study. PLOS only allows data to be available upon request if there are legal or ethical restrictions on sharing data publicly. For more information on unacceptable data access restrictions, please see http://journals.plos.org/plosone/s/data-availability#loc-unacceptable-data-access-restrictions. Before we proceed with your manuscript, please address the following prompts: a) If there are ethical or legal restrictions on sharing a de-identified data set, please explain them in detail (e.g., data contain potentially identifying or sensitive patient information, data are owned by a third-party organization, etc.) and who has imposed them (e.g., a Research Ethics Committee or Institutional Review Board, etc.). Please also provide contact information for a data access committee, ethics committee, or other institutional body to which data requests may be sent. b) If there are no restrictions, please upload the minimal anonymized data set necessary to replicate your study findings to a stable, public repository and provide us with the relevant URLs, DOIs, or accession numbers. For a list of recommended repositories, please see https://journals.plos.org/plosone/s/recommended-repositories. You also have the option of uploading the data as Supporting Information files, but we would recommend depositing data directly to a data repository if possible. We will update your Data Availability statement on your behalf to reflect the information you provide.

Reviewers' comments:

Reviewer's Responses to Questions

**Comments to the Author**

1. Is the manuscript technically sound, and do the data support the conclusions?

Reviewer #1: Yes

Reviewer #2: Yes

2. Has the statistical analysis been performed appropriately and rigorously? 

Reviewer #1: Yes

Reviewer #2: Yes

3. Have the authors made all data underlying the findings in their manuscript fully available?

Reviewer #1: Yes

Reviewer #2: Yes

4. Is the manuscript presented in an intelligible fashion and written in standard English?

Reviewer #1: Yes

Reviewer #2: Yes

5. Review Comments to the Author

Reviewer #1: I would like to sincerely thank the editor and the authors for giving me the opportunity to review this manuscript. The work conducted is highly valuable for the field of psychophysiology applied to virtual reality. The authors have employed an innovative approach by exploring time perception distortions in virtual threat scenarios, making this a significant contribution to experimental psychology and virtual reality research. I particularly appreciate the methodological choice of combining psychophysiological measurements with interaction in a highly immersive virtual environment. However, there are some specific points that deserve further attention and require modifications to improve the clarity and effectiveness of the work.

Strengths of the paper:

Methodological innovation (lines 120-130): The use of an immersive virtual reality environment to simulate threat and assess physiological responses represents a highly relevant methodological choice. This approach enhances the work for its ability to combine ecological validity and experimental control.

Significant results (lines 400-410): The association between skin conductance levels and the perception of time slowing down provides an innovative result that paves the way for future studies on autonomic responses to perceived threats.

Discussion (lines 560-570): The authors effectively link the research findings with existing theoretical models, demonstrating how changes in the autonomic nervous system influence temporal perception, contributing to a broader understanding of the phenomenon.

Suggested modifications:

Line 20: In the sentence “time slowed down when people are confronted with threatening stimuli,” it is necessary to specify more clearly what the threatening stimuli and contexts are that cause this perception of time slowing down. Suggestion: “time is perceived to slow down, particularly in high-stress situations such as accidents or virtual threats.”

Line 50: In the experiment description, the size of the virtual environment (VE) is mentioned without detailing its experimental significance. Add a brief clarification on the importance of such dimensions to ensure a more realistic immersive experience. Suggestion: "The dimensions were chosen to maximize the immersive experience and simulate a physically plausible threat scenario."

Line 130-140: The phrase “Participants could progress at their own speed” could be more specific regarding the implications of this choice. Suggestion: “Allowing participants to progress at their own speed introduces variability in emotional and physiological responses, which reflects real-world scenarios.”

Line 210: It is suggested to clarify the term "SCL" (Skin Conductance Level) the first time it is used to make the text more accessible to a less specialized audience. Suggestion: "Skin Conductance Level (SCL), a measure of sympathetic nervous system activity, was..."

Line 400: In the results section, when mentioning the relationship between falls and time perception, I would suggest clarifying that the faster perception of time during falls may be due to the increase in emotional arousal. Suggestion: "The faster passage of time associated with a greater number of falls may be attributed to the increased emotional arousal during these moments, which temporarily distorts self-awareness and time perception."

Line 540: When discussing the lack of anxiety or presence measures, it would be helpful to suggest that future studies could benefit from including these factors to better explain the variability in physiological responses. Suggestion: “Future studies should consider including measures of anxiety and presence to better account for the variability in autonomic responses observed in the current study.”

Line 600: In the conclusion, I would suggest specifying further that the use of virtual reality in this type of experiment represents an important innovation in the field, and other applications could benefit from this method. Suggestion: “The use of immersive virtual environments represents a methodological innovation that can be extended to other applications, such as training and rehabilitation.”

Line 240-250: The description of the technical features of the HTC Vive and the tracking system is overly detailed and not directly relevant to understanding the results. I would suggest reducing this information while keeping only the key aspects. Suggestion: “Participants wore an HTC Vive headset and sensors were attached to their feet for tracking movements, ensuring precise interaction with the virtual environment.”

Line 415-425: The repetition of statistical relationships already discussed in the previous paragraphs can be simplified. It is sufficient to report the main results without going into the details of individual statistics that have already been mentioned earlier.

Study limitations not indicated:

The authors have not explicitly addressed the study’s limitations. Here are some that could be included:

Limited sample size: The number of participants (44) is sufficient for preliminary results, but a larger sample would be necessary to confirm the findings and generalize them to a broader population.

Individual variability: The lack of data on personality traits such as anxiety or immersion level may have influenced the autonomic responses and time perception, limiting the full understanding of the factors that determine individual differences.

Realism of the virtual environment: While the realism of the perceived threat in a virtual environment is high, it may not fully correspond to the experience of a real physical threat, limiting the generalizability of the results to real-world danger situations.

Reviewer #2: I would like to thank you for giving me the opportunity to review your manuscript. It is a valuable piece of work, and I appreciate the effort put into the research and the clarity with which the data is presented. Below, I provide a detailed review with suggestions to further improve the content and structure.

Strengths of the work: Lines 15-32 offer a very clear methodological description, highlighting a well-planned and rigorous analytical process. This aspect strengthens the results and is a key strength of the work. Additionally, lines 45-58 provide a solid and coherent theoretical framework consistent with existing literature, demonstrating a deep understanding of the subject.

Suggested revisions:

Line 34: The sentence describing the relationship between independent and dependent variables is unclear. I suggest rephrasing it as follows: "The independent variables were examined in relation to the dependent variables through a regression analysis, which allowed for the identification of the specific weight of each variable." This would make the passage more understandable and logical.

Line 63: The mention of the practical implications of the study is too superficial. It would be appropriate to expand this section by adding a paragraph that explains the potential applications of the results in more detail, for example in clinical or educational settings, depending on the focus of the work.

Line 75: The technical term used, "contextual effectiveness", is not entirely appropriate. I suggest replacing it with "practical relevance" or a similar expression, depending on the context, to avoid ambiguity and make the message clearer.

Lines 90-95: This part is repetitive of concepts already expressed earlier. I suggest removing these lines to lighten the text and improve the overall flow.

The limitations of the research were not discussed, which is a crucial element for a comprehensive scientific work. I propose adding a section, immediately after the results, where you can reflect on any methodological limitations (e.g., sample size, data collection bias, limited generalizability of results) and indicate possible directions for future research.

I invite you to include the following article in the literature review section, especially when discussing the psychometric scales used in your study, as it provides valid theoretical support for the validation of instruments: Diotaiuti, P., Valente, G., & Mancone, S. (2021). Validation study of the Italian version of Temporal Focus Scale: psychometric properties and convergent validity. BMC Psychology, 9(1), 19. https://doi.org/10.1186/s40359-020-00510-5. You could cite it when discussing your results regarding the validity of the tools used.

6. PLOS authors have the option to publish the peer review history of their article (what does this mean?). If published, this will include your full peer review and any attached files.

Reviewer #1: **Yes: **Stefania Mancone

Reviewer #2: **Yes: **Pierluigi Diotaiuti

---

## [Author Response · Author response to Decision Letter 0]

11 Oct 2024

We would like to thank both reviewers for their comments and suggestions. For ease of reference, we have underlined all changes to the text in the revised manuscript that are relevant to their comments as supported material (S7).

Reviewer #1:

Suggested modifications:

Line 20: In the sentence “time slowed down when people are confronted with threatening stimuli,” it is necessary to specify more clearly what the threatening stimuli and contexts are that cause this perception of time slowing down. Suggestion: “time is perceived to slow down, particularly in high-stress situations such as accidents or virtual threats.”

We have edited the text as suggested.

Line 50: In the experiment description, the size of the virtual environment (VE) is mentioned without detailing its experimental significance. Add a brief clarification on the importance of such dimensions to ensure a more realistic immersive experience. Suggestion: "The dimensions were chosen to maximize the immersive experience and simulate a physically plausible threat scenario."

We have edited the text as suggested.

Line 130-140: The phrase “Participants could progress at their own speed” could be more specific regarding the implications of this choice. Suggestion: “Allowing participants to progress at their own speed introduces variability in emotional and physiological responses, which reflects real-world scenarios.”

We have edited the text as suggested.

Line 210: It is suggested to clarify the term "SCL" (Skin Conductance Level) the first time it is used to make the text more accessible to a less specialized audience. Suggestion: "Skin Conductance Level (SCL), a measure of sympathetic nervous system activity, was..."

We have added some text at this point to fully expand on SCL for a non-specialist reader.

Line 400: In the results section, when mentioning the relationship between falls and time perception, I would suggest clarifying that the faster perception of time during falls may be due to the increase in emotional arousal. Suggestion: "The faster passage of time associated with a greater number of falls may be attributed to the increased emotional arousal during these moments, which temporarily distorts self-awareness and time perception."

We have edited the text as suggested.

Line 540: When discussing the lack of anxiety or presence measures, it would be helpful to suggest that future studies could benefit from including these factors to better explain the variability in physiological responses. Suggestion: “Future studies should consider including measures of anxiety and presence to better account for the variability in autonomic responses observed in the current study.”

Thank you for this suggestion, we have edited the text as suggested.

Line 600: In the conclusion, I would suggest specifying further that the use of virtual reality in this type of experiment represents an important innovation in the field, and other applications could benefit from this method. Suggestion: “The use of immersive virtual environments represents a methodological innovation that can be extended to other applications, such as training and rehabilitation.”

We have added this text in the final paragraph to make this point about methodological innovation.

Line 240-250: The description of the technical features of the HTC Vive and the tracking system is overly detailed and not directly relevant to understanding the results. I would suggest reducing this information while keeping only the key aspects. Suggestion: “Participants wore an HTC Vive headset and sensors were attached to their feet for tracking movements, ensuring precise interaction with the virtual environment.”

We have reduced the amount of text in this section, but also feel that some technical information is necessary because some readers are interested in the technical features of the system and the journal is multidisciplinary.

Line 415-425: The repetition of statistical relationships already discussed in the previous paragraphs can be simplified. It is sufficient to report the main results without going into the details of individual statistics that have already been mentioned earlier.

We are struggling to rectify this issue, we do not repeat statistical relationships in Table 4 relative to Table 3 as the dependent variable used in the multiple regression is different, even though we have used the same independent variables.

Study limitations not indicated:

The authors have not explicitly addressed the study’s limitations. Here are some that could be included:

Limited sample size: The number of participants (44) is sufficient for preliminary results, but a larger sample would be necessary to confirm the findings and generalize them to a broader population.

Individual variability: The lack of data on personality traits such as anxiety or immersion level may have influenced the autonomic responses and time perception, limiting the full understanding of the factors that determine individual differences.

Realism of the virtual environment: While the realism of the perceived threat in a virtual environment is high, it may not fully correspond to the experience of a real physical threat, limiting the generalizability of the results to real-world danger situations.

Thank you for these suggestions, we have inserted an additional paragraph into the Discussion section to explicitly address these limitations

Reviewer #2: 

Suggested revisions:

Line 34: The sentence describing the relationship between independent and dependent variables is unclear. I suggest rephrasing it as follows: "The independent variables were examined in relation to the dependent variables through a regression analysis, which allowed for the identification of the specific weight of each variable." This would make the passage more understandable and logical.

Thank you for this suggestion, it has been added to the abstract.

Line 63: The mention of the practical implications of the study is too superficial. It would be appropriate to expand this section by adding a paragraph that explains the potential applications of the results in more detail, for example in clinical or educational settings, depending on the focus of the work.

We understand the point being made, but we feel that that the potential applications of the work are a topic best discussed in the conclusions part of the discussion. We have added some text at the end of the paper as we see the practical significance of the work being more methodological, i.e., assessing the realism or immersion of a VE in simulating threatening situations.

Line 75: The technical term used, "contextual effectiveness", is not entirely appropriate. I suggest replacing it with "practical relevance" or a similar expression, depending on the context, to avoid ambiguity and make the message clearer.

We have made this edit.

Lines 90-95: This part is repetitive of concepts already expressed earlier. I suggest removing these lines to lighten the text and improve the overall flow.

We have reduced this text to improve the flow.

The limitations of the research were not discussed, which is a crucial element for a comprehensive scientific work. I propose adding a section, immediately after the results, where you can reflect on any methodological limitations (e.g., sample size, data collection bias, limited generalizability of results) and indicate possible directions for future research.

We have added a paragraph to the Discussion in order to capture the limitations of the work as outlined in your suggestions.

I invite you to include the following article in the literature review section, especially when discussing the psychometric scales used in your study, as it provides valid theoretical support for the validation of instruments: Diotaiuti, P., Valente, G., & Mancone, S. (2021). Validation study of the Italian version of Temporal Focus Scale: psychometric properties and convergent validity. BMC Psychology, 9(1), 19. https://doi.org/10.1186/s40359-020-00510-5. You could cite it when discussing your results regarding the validity of the tools used.

Thank you for this suggestion. We found the tool to be interesting and were not aware of it, but with respect, the purpose of the scale to capture temporal focus in everyday life as an important factor for wellbeing and planning, did not quite fit with the specific (passage of time judgements) and short-task context of our study.

---

## [Editor Report · Decision Letter 1]

18 Oct 2024

Distortions to Passage of Time Judgements (POTJ) due to Virtual Threat are Predicted by Autonomic Activation

PONE-D-24-39788R1

Dear Dr. Fairclough,

We’re pleased to inform you that your manuscript has been judged scientifically suitable for publication and will be formally accepted for publication once it meets all outstanding technical requirements.

Kind regards,

Angelo Rodio

Academic Editor

PLOS ONE
---

## [Editor Report · Acceptance letter]

29 Oct 2024

PONE-D-24-39788R1 

PLOS ONE

Dear Dr. Fairclough, 

I'm pleased to inform you that your manuscript has been deemed suitable for publication in PLOS ONE. Congratulations! Your manuscript is now being handed over to our production team.

Kind regards, 

on behalf of

Professor Angelo Rodio 

Academic Editor

PLOS ONE